

# Slower EEG alpha generation, synchronization and "flow"—possible biomarkers of cognitive impairment and neuropathology of minor stroke

Jelena Petrovic[1], Vuk Milosevic[2], Miroslava Zivkovic[2], Dragan Stojanov[3], Olga Milojkovic[4], Aleksandar Kalauzi[5] and Jasna Saponjic[1]

[1] Department of Neurobiology, Institute for Biological Research—Sinisa Stankovic, University of Belgrade, Belgrade, Serbia
[2] Clinic of Neurology, Clinical Center Nis, Nis, Serbia
[3] Institute of Radiology, Clinical Center Nis, Nis, Serbia
[4] Clinic for Mental Health Protection, Clinical Center Nis, Nis, Serbia
[5] Department for Life Sciences, Institute for Multidisciplinary Research, University of Belgrade, Belgrade, Serbia

Corresponding author
Jelena Petrovic,
jelena.petrovic@ibiss.bg.ac.rs

## ABSTRACT

**Background**. We investigated EEG rhythms, particularly alpha activity, and their relationship to post-stroke neuropathology and cognitive functions in the subacute and chronic stages of minor strokes.

**Methods**. We included 10 patients with right middle cerebral artery (MCA) ischemic strokes and 11 healthy controls. All the assessments of stroke patients were done both in the subacute and chronic stages. Neurological impairment was measured using the National Institute of Health Stroke Scale (NIHSS), whereas cognitive functions were assessed using the Montreal Cognitive Assessment (MoCA) and MoCA memory index (MoCA-MIS). The EEG was recorded using a 19 channel EEG system with standard EEG electrode placement. In particular, we analyzed the EEGs derived from the four lateral frontal (F3, F7, F4, F8), and corresponding lateral posterior (P3, P4, T5, T6) electrodes. Quantitative EEG analysis included: the group FFT spectra, the weighted average of alpha frequency (αAVG), the group probability density distributions of all conventional EEG frequency band relative amplitudes (EEG microstructure), the inter- and intra-hemispheric coherences, and the topographic distribution of alpha carrier frequency phase potentials (PPs). Statistical analysis was done using a Kruskal–Wallis ANOVA with a *post-hoc* Mann–Whitney *U* two-tailed test, and Spearman's correlation.

**Results**. We demonstrated transient cognitive impairment alongside a slower alpha frequency (αAVG) in the subacute right MCA stroke patients vs. the controls. This slower alpha frequency showed no amplitude change, but was highly synchronized intra-hemispherically, overlying the ipsi-lesional hemisphere, and inter-hemispherically, overlying the frontal cortex. In addition, the disturbances in EEG alpha activity in subacute stroke patients were expressed as a decrease in alpha PPs over the frontal cortex and an altered "alpha flow", indicating the sustained augmentation of inter-hemispheric interactions. Although the stroke induced slower alpha was a transient phenomenon, the increased alpha intra-hemispheric synchronization, overlying the ipsi-lesional hemisphere, the increased alpha F3–F4 inter-hemispheric synchronization, the delayed alpha waves, and the newly established inter-hemispheric "alpha flow"

within the frontal cortex, remained as a permanent consequence of the minor stroke. This newly established frontal inter-hemispheric "alpha flow" represented a permanent consequence of the "hidden" stroke neuropathology, despite the fact that cognitive impairment has been returned to the control values. All the detected permanent changes at the EEG level with no cognitive impairment after a minor stroke could be a way for the brain to compensate for the lesion and restore the lost function.

**Discussion**. Our study indicates slower EEG alpha generation, synchronization and "flow" as potential biomarkers of cognitive impairment onset and/or compensatory post-stroke re-organizational processes.

# INTRODUCTION

Stroke is a common neurological disease with variable outcomes regarding the level of disabilities and mortality caused (*Sturm et al., 2002*). In most cases (80–90%) a stroke occurs as a result of reduced blood flow, due to an obstructed blood vessels (*Grefkes & Ward, 2014*), causing deficits in a number of neurological domains, including cognitive impairment (*Cramer, 2008*). The initial and sudden cognitive decline after a stroke is characterized by partial recovery within a few months (*Thiel et al., 2014*). However, cognitive assessment at least three months after a stroke shows more stable, long-lasting cognitive impairment. Although varying between countries, races and diagnostic criteria, the prevalence of post-stroke cognitive impairment in stroke survivors is high. Between 7 and 41% of ischemic stroke patients have dementia three to 12 months after their stroke and more than 80% show cognitive impairment in one or multiple domains which does not fulfil the criteria for dementia (*Jokinen et al., 2015*; *Mijajlović et al., 2017*; *Pendlebury & Rothwell, 2009*; *Sun, Tan & Yu, 2014*). Cognitive impairment significantly influences the independent functioning and quality of life of stroke patients. However, given the severity of the physical disabilities involved, post-stroke cognitive impairment is often neglected (*Mijajlović et al., 2017*).

The post-stroke decline in general cognitive status occurs as a consequence of diverse cortical and subcortical strokes, with resulting neurocognitive deficits, reflecting the extent and location of the lesions (*Sheorajpanday et al., 2014*). The degree of deviation from normal cognition in stroke patients is highly variable in terms of both type and severity, since different domains, including executive function, attention, memory, visuospatial abilities, language and orientation, may be affected to varying degrees (*Kalaria, Akinyemi & Ihara, 2016*). All of these impairments, particularly subtle cognitive dysfunction, might easily go unnoticed without appropriate screening. The most commonly used method for measuring cognitive impairment is a neuropsychological assessment which employs brief screening tests, such as the Mini Mental State Examination (MMSE) (*Folstein, Folstein & McHugh, 1975*), and the Montreal Cognitive Assessment (MoCA) (*Nasreddine et al., 2005*). Although widely accepted for cognitive assessment, none of these tests are
specifically designed for post-stroke cognitive screening. Therefore, despite their ability to identify cognitive dysfunction, these tests have limitations regarding their sensitivity in detecting all cases of cognitive impairment, their selectivity in ruling out those without cognitive impairment, and finally, the types of stroke patients that can be assessed (*Schleiger et al., 2014*).

EEG is a low-cost, non-invasive imaging technique that has been used in clinical practice for decades. Having excellent temporal resolution, EEG provides a direct measurement of the cerebral functional status, with subtle EEG changes reflecting underlying pathophysiological processes (*Borich et al., 2015*). EEG abnormalities are typical manifestations of an ischemic stroke. Following cerebral blood flow reduction, the metabolic and electrical activities of cortical neurons are altered (*George & Steinberg, 2015*; *Hossmann, 1994*). These alterations can be observed as specific EEG patterns over the ischemic area, coming from the attenuation of faster (alpha and beta) and the augmentation of slower (delta and theta) frequency bands (*Jordan, 2004*; *Nagata et al., 1989*). Apart from being very sensitive in the detection of cerebral ischemia, EEG can also be used as a diagnostic and prognostic tool, or to track therapy during stroke recovery (*Burghaus et al., 2013*; *Finnigan & Van Putten, 2013*; *Finnigan, Wong & Read, 2016*; *Foreman & Claassen, 2012*). Namely, according to recent studies, the quantitative measures of specific EEG features (qEEG) are even more sensitive than raw EEG, and well correlated with stroke severity and radiographic findings (*Finnigan et al., 2004*; *Sheorajpanday et al., 2011*), response to treatment such as thrombolytic therapy (*Finnigan, Rose & Chalk, 2006*), and to functional outcomes (*Burghaus et al., 2013*; *Finnigan et al., 2004*).

In addition, many studies demonstrated that some qEEG indices are also sensitive to cognitive function (*Klimesch, 2012*) and mild cognitive impairment or dementia (*Dubovik et al., 2013*; *Schleiger et al., 2014*; *Schleiger et al., 2017*; *Song et al., 2015*). In line with the general view regarding cognitive and memory performance (*Klimesch, 1999*), these studies indicated that the alpha, as well as theta frequency, could be of interest for the screening of post-stroke cognitive impairment (*Dubovik et al., 2013*; *Schleiger et al., 2014*; *Schleiger et al., 2017*; *Song et al., 2015*). In contrast to relative theta power, whether global, as demonstrated by *Song et al. (2015)*, or posterior, computed from three electrodes, as shown in *Schleiger et al. (2017)*, the qEEG assessments revealed global alpha power (*Schleiger et al., 2014*), EEG alpha band synchrony (*Dubovik et al., 2013*), as well as a lower peak alpha frequency (*Schleiger et al., 2017*) to be informative as regards post-stroke cognitive impairment.

Since cortical activity varies during each stage of a stroke, the assessment of a stroke by EEG measurements has proven to be an essential monitoring standard for identifying spatiotemporal markers of cortical activity related to injury, the post-stroke re-organizational processes, and recovery (*Iyer, 2017*). The initial slowing of brain EEG activity in acute strokes could be characterized by analyzing the power of slower relative to faster EEG activities, such as the delta/alpha ratio (*Finnigan, Wong & Read, 2016*; *Schleiger et al., 2014*). As the stroke progresses to the subacute and chronic stages, functional connectivity measures, such as the phase synchrony within a specific frequency band like alpha (*Dubovik et al., 2012*; *Westlake et al., 2012*), become more important for studying re-organizational processes, brain plasticity, and functional recovery (*Iyer, 2017*).

All the abovementioned findings suggest that EEG measurements can be employed to assess the general status of the brain and to complement neuropsychological testing when studying cognitive impairment following a stroke, potentially resulting in specific qEEG assesments that could be used as biomarkers of cognitive impairment and/or compensatory post-stroke re-organizational processes. Therefore, the primary aim of this study is to investigate the alterations in EEG rhythms, and particularly in alpha activity, as a reflection of the underlying post-stroke neuropathology. Our secondary aim is to relate these EEG alterations to post-stroke re-organizational processes and cognitive functions in the subacute and chronic stages of a minor ischemic stroke vs. the controls.

## MATERIALS AND METHODS

### Subjects

This research was conducted at the Clinic of Neurology of the Clinical Centre in Nis, Serbia. We included 21 subjects: 10 ischemic stroke patients and 11 healthy controls. One of the stroke patients died during the follow-up period. The clinical diagnosis of stroke and its classification were determined by a detailed patient history of symptoms, a neurological examination, and computed tomography (CT) and/or magnetic resonance imaging (MRI). Patients were assessed $9.70 \pm 2.50$ days following the onset of their stroke (the subacute stage for the first assessment), as well as $13.67 \pm 5.85$ months after their stroke (the chronic stage for the follow-up assessment). Neurological impairment was measured using the National Institute of Health Stroke Scale (NIHSS) on admission and at the time of both the first assessment and the follow-up assessment. Cognitive function was assessed by the Montreal Cognitive Assessment (MoCA) (*Nasreddine et al., 2005*) and the MoCA memory index score (MoCA-MIS) (*Julayanont et al., 2014*) both in the subacute stage and during the follow up. The Edinburgh handedness inventory-short form (EHI-SF) with four claims and a five-point response scale was used for assessing hand dominance (*Veale, 2014*).

The inclusion criteria for the minor stroke patients were the following: the first ever stroke, a confirmed ischemic stroke in the distribution of right middle cerebral artery (MCA), an NIHSS of four or less, the absence of aphasia, right hand dominance (EHI-SF score > 60) and the absence of other neurological or psychiatric diseases. The control subjects were volunteers with no history of neurological or psychiatric disorders who were matched by age and years of formal education with the stroke group. We used the Informant Questionnaire on Cognitive Decline in the Elderly (IQCODE)—short form (*Jorm, 1994*) to identify and exclude patients with pre-stroke dementia (IQCODE score > 3). The study was carried out in accordance with the Declaration of Helsinki (The Code of Ethics of the World Medical Association), and approved by the Ethical Committee of the Medical Faculty of the University of Nis, Republic of Serbia (No 12-9808-2/1). All the participants gave informed consent.

### EEG data acquisition

EEG was recorded on the day of assessment, both in the subacute and the chronic stages of stroke, using a 19 channel NK-9100K EEG system (Nihon Kohden, Japan). We used standard EEG electrode placement and positioning according to the 10–20 International

System (Fp1, Fp2, F7, F3, F4, F8, T3, C3, C4, T4, T5, P3, P4, T6, O1, and O2). Recordings were referenced to (C3+C4)/2, the physical reference of the NK-9100K EEG system, with the ground electrode placed on the forehead. After conventional amplification and filtering (high pass 0.1 Hz, notch filter 50 Hz) the analog signals were digitized (250/s), and recorded for approximately 15 min in a resting, awake state with eyes closed.

## EEG data analysis

All the EEG recordings were visually inspected and discriminated using EEGLAB to eliminate any ocular, muscular, and other types of artefacts before further analysis. The final EEG signals, consisting of 10 min artefact-free periods, were analyzed in MATLAB R2011a, using software originally developed in MATLAB 6.5 (*Ciric et al., 2015*; *Ciric et al., 2016*; *Kalauzi, Vuckovic & Bojić, 2012*; *Lazic et al., 2015*; *Lazic et al., 2017*; *Petrovic et al., 2013a*; *Petrovic et al., 2013b*; *Petrovic et al., 2014*; *Saponjic et al., 2013*). We particularly analyzed the EEGs derived from the four lateral frontal (F3, F7, F4, F8), and corresponding lateral posterior (P3, P4, T5, T6) electrodes (*Schleiger et al., 2014*). After additional filtering (1–50 Hz band pass filter), we applied FFT algorithm to each individual recording, using a 10 s non-overlapping moving window resulting in 0.1 Hz frequency resolution (60 10 s Fourier epochs per each electrode), to calculate the relative EEG amplitudes of all the conventional frequency bands ($\delta = 1.1$–4 Hz; $\theta = 4.1$–8 Hz; $\alpha = 8.1$–13 Hz; $\beta = 13.1$–30 Hz; $\gamma = 30.1$–50 Hz).

Following the concept of "individual alpha frequency" (*Klimesch, 1999*), for each EEG recording and each electrode we calculated the weighted average of the alpha frequency ($\alpha$AVG), using the alpha amplitude as a weighting factor:

$$\alpha AVG = \frac{\sum_f (Amp(f) x f)}{\sum_f Amp(f)}$$

where $f$ is a specific frequency ranging across the alpha frequency band and $Amp(f)$ is the amplitude at $f$ (*Hooper, 2005*).

In addition, to analyze the EEG amplitude changes (the EEG microstructure), we calculated the group probability density distributions of all the conventional EEG frequency band relative amplitudes using the Probability Density Estimate (PDE) routine supplied with MATLAB R2011a. As in our previous studies, in order to eliminate any influence from absolute signal amplitude variations on the recordings, we calculated the relative Fourier amplitudes (*Ciric et al., 2015*; *Ciric et al., 2016*; *Lazic et al., 2015*; *Lazic et al., 2017*; *Petrovic et al., 2013a*; *Petrovic et al., 2013b*; *Petrovic et al., 2014*; *Saponjic et al., 2013*):

$$(RA)_b = \frac{\sum_b Amp}{\sum_{tot} Amp}, b = \{\delta, \theta, \alpha, \beta, \gamma\}.$$

For each electrode and each frequency band, PDE analysis was performed on the ensembles of relative amplitudes by pooling the measured values from all subjects belonging to the corresponding group (the control, the subacute stage of the stroke or the chronic stage of the stroke). For the statistical analysis of PDEs the relative amplitude means were calculated for each 2 min.

We also calculated the inter-hemispheric and intra-hemispheric coherences by using the "mscohere" routine of the MATLAB R2011a Signal Processing Toolbox, which computes the magnitude squared coherence between the signals $x$ and $y$ as:

$$C_{(xy)}(f) = \frac{|P_{xy}(f)|^2}{P_{xx}(f)P_{yy}(f)}$$

where $P_{xy}(f)$ is the cross spectrum of $x$ and $y$, while $P_{xx}(f)$ and $P_{yy}(f)$ denote the power spectra of the two signals. For this purpose, the individual EEG signals, derived from the corresponding electrodes, were concatenated and pooled within each group. For each electrode pair, a coherence value of between 0 and 1 was calculated for every 10 s and each frequency point (at a 0.1 Hz frequency resolution), within the overall 0.1–50 Hz range. Then the values within each conventional frequency band were averaged for each spectrum, and their means were calculated from the collection of all the available spectra. We calculated the group mean coherence values for each 2 min, for each frequency band, each pair of electrodes, and each group.

We also analyzed the topographic distribution of the alpha carrier frequency phase potentials (PPs) through changes in their carrier frequency phase shifts (PSs; *Kalauzi, Kesic & Saponjic, 2009*; *Kalauzi, Vuckovic & Bojić, 2012*). For this particular analysis, the individual EEG signals, derived from all 19 electrodes, were re-referenced offline to the common average reference (the average of all 19 electrodes). To be specific, for each pair of channels, their EEG recordings were treated as signals with variable amplitudes, but with the same alpha frequency (model details can be found in *Kalauzi, Vuckovic & Bojić, 2012*, Appendix B). Since alpha frequency is expected to be different between individuals ("individual alpha frequency", *Klimesch, 1999*), the alpha frequency range limits used for PP analysis were determined individually, based on the previously determined $\alpha$AVG, following the natural limits of the alpha peak position within each spectrum. For each subject and each pair of electrodes we calculated an alpha carrier frequency PS and alpha carrier frequency PP derived from the corresponding Fourier component PSs. Finally, we calculated the group alpha carrier frequency PPs (for the control, the subacute stage of the stroke and the chronic stage of the stroke), and the group alpha carrier frequency PP differences (for the subacute stage of the stroke vs. the control, the chronic stage of the stroke vs. the control, and the chronic stage of stroke vs. the subacute stage of stroke) for each electrode. All the calculations (the alpha carrier frequency PSs, the alpha carrier frequency PPs, the group mean alpha carrier frequency PPs and the group alpha carrier frequency PP differences) were done using angular arithmetic (*Kalauzi, Vuckovic & Bojić, 2012*, Appendix A, C, E). Since alpha carrier frequency PP is an integral phase measure of a particular EEG channel in the alpha range, incorporating all alpha Fourier components, it is possible to compare and subtract carrier frequency PP values even if the $\alpha$AVG frequency differs between groups, as was the case in this study. The final results of PP analysis are presented as circles over corresponding electrodes, with diameters (small or large) referring to the PP absolute values, and line types (solid or dashed) referring to the PP signs (positive or negative). To be clear, since the PP calculation is based on angular variables, small circles correspond to abs (PP) $\approx 0°$, while large circles correspond to abs (PP) $\approx \pm 180°$ (with a

**Table 1** Demographic and clinical characteristics of stroke patients and healthy controls.

|  | Control | Subacute stroke | Chronic stroke |
|---|---|---|---|
| Number | 11 | 10 | 9 |
| Gender (m/f) | 5/6 | 6/4 | 5/4 |
| Age (years) | $60.64 \pm 9.91$ | $63.80 \pm 7.00$ | $63.89 \pm 6.83$ |
| NIHSS (on admission) | / | $8.20 \pm 3.05$ | / |
| NIHSS | / | $2.50 \pm 0.85$ | $2.11 \pm 0.78$ |
| Education (years) | $12.45 \pm 3.11$ | $9.80 \pm 2.70$ | / |
| MoCA score | $25.64 \pm 3.20$ | **$20.80 \pm 5.32$** | $23.44 \pm 4.98$ |
| MoCA-MIS | $11.91 \pm 2.98$ | **$8.80 \pm 3.52$** | $10.22 \pm 3.35$ |

**Notes.**
Bold numbers indicate statistically significant values at $p \leq 0.05$.

solid line for positive and a dashed line for negative PP values). Therefore, due to their angular nature, the large diameter circles represent similar PP values, regardless of the sign (e.g., PP $= 178°$ and PP $= -179°$ should be treated as close to each other since their actual difference is not $357°$, but rather $3°$). In this way all channels can be organized in a series (order) based on their PP values: those channels leading in phase to all others will have the highest (large positive) PP values, while the channels with the lowest (large negative) PP values would be the one following all the others (*Kalauzi, Vuckovic & Bojić, 2012*). Based on these results, we have also drawn a topographic pattern of ''alpha flow'' in the control group, and the subacute and chronic stage of the stroke, by following the phase potential gradient (from the highest to the lowest PP values).

## Statistical analysis

All the values are presented as means $\pm$ SDs. The statistical analysis was done using the Kruskal–Wallis ANOVA with a *post-hoc* Mann–Whitney $U$ two-tailed test, and Spearman's correlation. In all cases the difference was considered significant for $p \leq 0.05$.

## RESULTS

### Clinical and demographic assessments

Table 1 summarizes the relevant demographic and clinical data of the control group and the stroke patients. There were six male and four female subjects among the stroke patients and five male and six female subjects in the control group. The age difference between stroke patients ($63.80 \pm 7.00$ in the subacute stage and $63.89 \pm 6.83$ in the chronic stage) and the healthy controls ($60.64 \pm 9.91$) was not statistically significant ($z \geq -0.78; p \geq 0.44$). There was no statistically significant difference in the years of education between the two groups ($z = -1.52; p = 0.13$). The average NIHSS was $8.20 \pm 3.05$ on admission, $2.50 \pm 0.85$ at the time of the first assessment, and $2.11 \pm 0.78$ at a time of the follow-up assessment.

All the patients in the stroke group had their stroke in the vascular territory of the right MCA. The average lesion volume measured by the ABC/2 method was $9.34 \pm 6.49$ cm$^3$. The ischemic stroke lesion volumes and the affected structures for each patient are shown in Table 2.

**Table 2  Patient's stroke details (Stroke neuropathology: the volume of lesions and the brain structures affected).**

| Patient no. | Gender (m/f) | Imaging modality | Volume (cm³) | Subcortical structures affected | Cortical structres affected |
|---|---|---|---|---|---|
| 1 | m | CT | 18.90 | Putamen | Insular cortex |
| 2 | m | CT | 9.11 | Putamen | |
| 3 | m | CT | 0.90 | | Insular cortex |
| 4 | f | CT | 13.19 | Putamen | |
| 5 | m | CT | 9.04 | | Superior temporal gyrus |
| 6 | f | CT | 14.04 | Putamen | Insular cortex, superior temporal gyrus |
| 7 | f | CT | 0.96 | Putamen | |
| 8 | m | MRI | 10.20 | Putamen, globus pallidus | |
| 9 | m | CT | 15.84 | | Transverse temporal gyrus, supramarginal gyrus |
| 10 | f | MRI | 1.20 | | Precentral gyrus, postcentral gyrus |

As indicated in Table 1, MoCA scores in stroke patients during the subacute stage were significantly lower than in the healthy controls ($20.80 \pm 5.32$ vs. $25.64 \pm 3.20$; $z = -2.12$; $p = 0.03$), in contrast with the chronic stage of the stroke where these MoCA score differences were not statistically significant in stroke patients vs. controls ($23.44 \pm 4.98$ vs. $25.64 \pm 3.20$; $z = -0.88$; $p = 0.38$).

In addition, there was a significant difference between the subacute stroke patients and the control group concerning MoCA-MIS (Table 1, $8.80 \pm 3.52$ vs. $11.91 \pm 2.98$; $z = -1.98$; $p = 0.05$), whereas in the chronic stroke patients, the MoCA-MIS score returned to the control values (Table 1, $10.22 \pm 3.35$ vs. $11.91 \pm 2.98$; $z = -1.15$; $p = 0.25$).

### EEG alpha activity following a minor stroke

For all the analyzed frontal and posterior EEGs, the group mean relative amplitude EEG spectra showed prominent amplitude peaks in the alpha frequency range in all groups, but with the alpha peaks shifted toward lower frequencies in stroke patients, particularly in the subacute stage (Fig. 1). Further analysis of $\alpha$AVG frequency revealed a generalized decrease in $\alpha$AVG following a stroke. In particular, and in contrast to the $\alpha$AVG in the control group that was in the range of 10.38–10.54 Hz, the subacute stroke group had an $\alpha$AVG in the range of 10.11–10.23 Hz, whereas the chronic stroke group had an $\alpha$AVG in the range of 10.22–10.38 Hz.

A statistically significant slower alpha frequency in stroke patients vs. controls was demonstrated only for the subacute stage at F3, F4, F7, F8, P3 and P4 ($\chi^2 \geq 6.11$; $p \leq 0.05$; $z \geq -3.32$; $p \leq 0.03$), but with no change in the $\alpha$AVG at T5 and T6 ($\chi^2 \geq 3.09$; $p \geq 0.18$; $z \geq -1.66$; $p \geq 0.10$). However, all the stroke induced alpha frequency changes returned to the control values ($z \geq -1.90$; $p \geq 0.06$) in the chronic stage.

In our study we did not demonstrate the amplitude change of the slower alpha in subacute and chronic stroke patients vs. controls ($\chi^2 \geq 1.44$; $p \geq 0.14$; $z \geq -1.72$; $p \geq 0.09$; data not shown), but we have shown the delta, theta, and beta amplitude alterations (Fig. 2; $\chi^2 \geq 6.61$; $p \leq 0.04$). The frontal cortex exhibited a general beta amplitude decrease in both subacute and chronic stroke patients vs. controls (Figs. 2I–2L, beta; $\chi^2 \geq 15.21$; $p = 10^{-4}$;

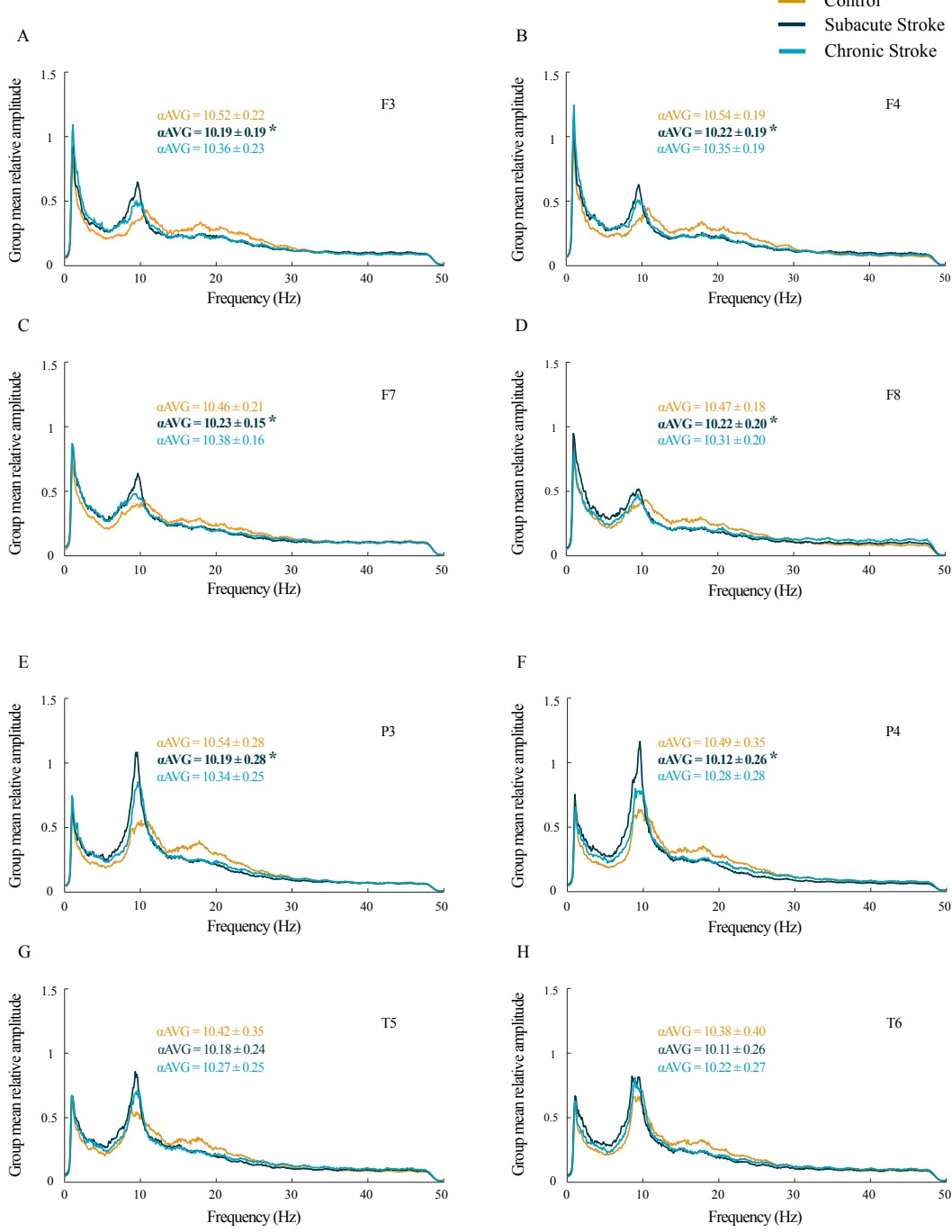

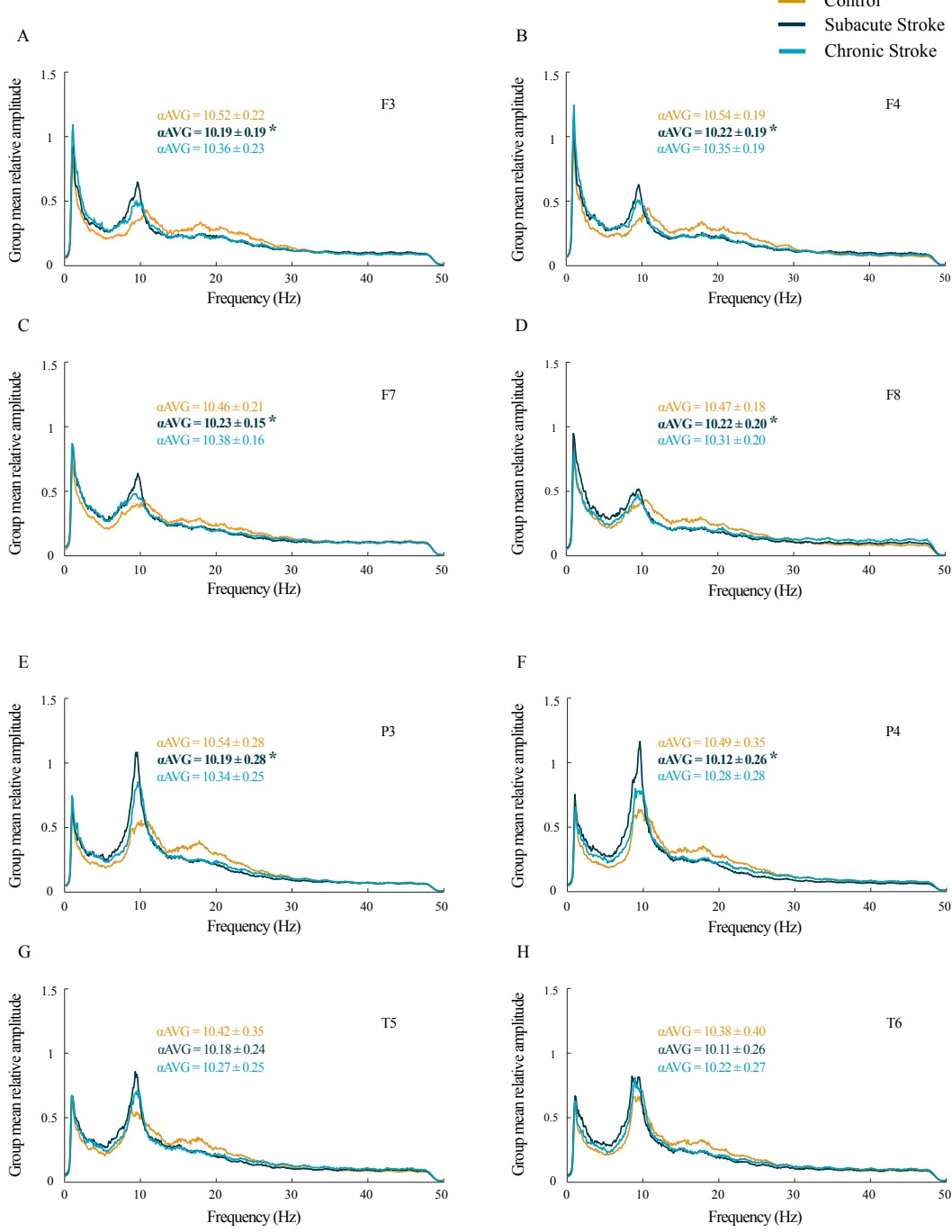

**Figure 1 Alterations in alpha frequency following a minor stroke.** Group mean relative amplitude spectra of the frontal (A–D), parietal and temporal (E–H) EEG channels in the subacute ($n = 10$) and chronic ($n = 9$) stages of stroke patients vs. healthy controls ($n = 11$). We have demonstrated statistically significant slower alpha ($\alpha$AVG) at F3, F4, F7, F8, P3 and P4 in the subacute stroke patients vs. the controls ($p \leq 0.03$). In the chronic stage, all stroke induced alpha frequency changes returned to the control values ($p \geq 0.06$). Bold numbers and asterisks indicate a statistically significant decrease in $\alpha$AVG following a stroke.

$z \geq -4.08$; $p \leq 0.01$). This beta amplitude decrease was followed by a long-lasting theta amplitude increase across all the frontal and posterior channels (Figs. 2E–2H and 2Q–2T, theta; $\chi^2 \geq 7.86$; $p \leq 0.02$; $z \geq -5.64$; $p \leq 0.03$), with the exception of F8 and T5 where the theta amplitude returned to the control values in the chronic stage ($z \geq -1.50$; $p \geq 0.13$). Delta amplitude alterations were in evidence only in the posterior channels (Figs. 2M–2P, delta; $\chi^2 \geq 6.61$; $p \leq 0.04$): the delta amplitude increased at P3 in the chronic stage of stroke patients vs. controls ($z = -2.42$; $p = 0.02$) in contrast to a transient increase at T5 and T6 in the subacute stage ($z \geq -2.90$; $p \leq 0.01$).

### Inter-hemispheric and intra-hemispheric coherences following a minor stroke

Inter-hemispheric coherence analysis between the frontal electrodes, in both subacute and chronic stroke patients, revealed significantly increased coherences in the alpha frequency range in contrast to the control group (Figs. 3A and 3B; $\chi^2 \geq 6.56$; $p \leq 0.04$; $z \geq -3.24$; $p \leq 0.02$). This sustainably increased synchronization of slower alpha, caused by the stroke, was particularly expressed at the F3–F4 electrodes, and co-localized with the underlying neuropathology of the ischemic stroke (see Table 2). Conversely, the posterior electrodes expressed a generalized long-lasting desynchronization in all the frequency ranges (Figs. 3C and 3D; $\chi^2 \geq 6.87$; $p \leq 0.03$; $z \geq -4.82$; $p \leq 0.04$), but with no change in alpha coherences ($\chi^2 \geq 2.59$; $p \geq 0.14$; $z \geq 10.63$; $p \geq 0.10$) in either subacute or chronic stroke patients vs. the controls.

In addition, in the chronic stage we demonstrated increased theta and beta synchronizations at the F3–F4 electrodes (Fig. 3A; $\chi^2 \geq 7.67$; $p \leq 0.02$; $z \geq -2.84$; $p \leq 0.02$) alongside broad spectrum (delta, theta, beta and gamma) desynchronization at the frontal F7–F8 electrodes (Fig. 3B; $\chi^2 \geq 6.59$; $p \leq 0.04$; $z \geq -3.00$; $p \leq 0.05$).

The increased synchronization of slower alpha in stroke patients vs. the controls was also shown within the ipsi-lesional hemisphere (Figs. 4D–4F; $\chi^2 \geq 6.06$; $p \leq 0.05$). We demonstrated sustainably increased alpha coherences across the F4–F8, F4–T6 and F4–P4 electrodes during both stages of stroke vs. the controls ($z \geq -5.11$; $p \leq 0.04$). In addition, there were also inconsistent alterations in the delta, theta, beta and gamma coherences within the ipsi-lesional hemisphere in the subacute and chronic stages of stroke patients vs. the controls (Figs. 4D–4F; $\chi^2 \geq 6.84$; $p \leq 0.03$; $z \geq -4.24$; $p \leq 0.02$). Conversely, the contra-lesional hemisphere demonstrated an increased synchronization in the theta frequency range, but only in subacute stroke patients (Figs. 4A–4C; $\chi^2 \geq 6.49$; $p \leq 0.04$; $z \geq -4.51$; $p \leq 0.05$). This transient theta coherence augmentation was followed by inconsistent delta, alpha, beta and gamma coherence alterations in both the subacute and chronic stroke stages vs. the controls (Figs. 4A–4C; $\chi^2 \geq 9.08$; $p \leq 0.01$; $z \geq -4.86$; $p \leq 0.03$).

### Topography of αAVG phase potentials following a minor stroke

The group mean phase potentials (PPs) of the newly emerged slower alpha in subacute stroke patients, and the alpha in the chronic stroke patients and healthy controls (Fig. 1) with their corresponding PP differences are depicted in Fig. 5. We have shown that the frontal cortex of the stroke patients vs. the controls expressed the greatest and long-lasting

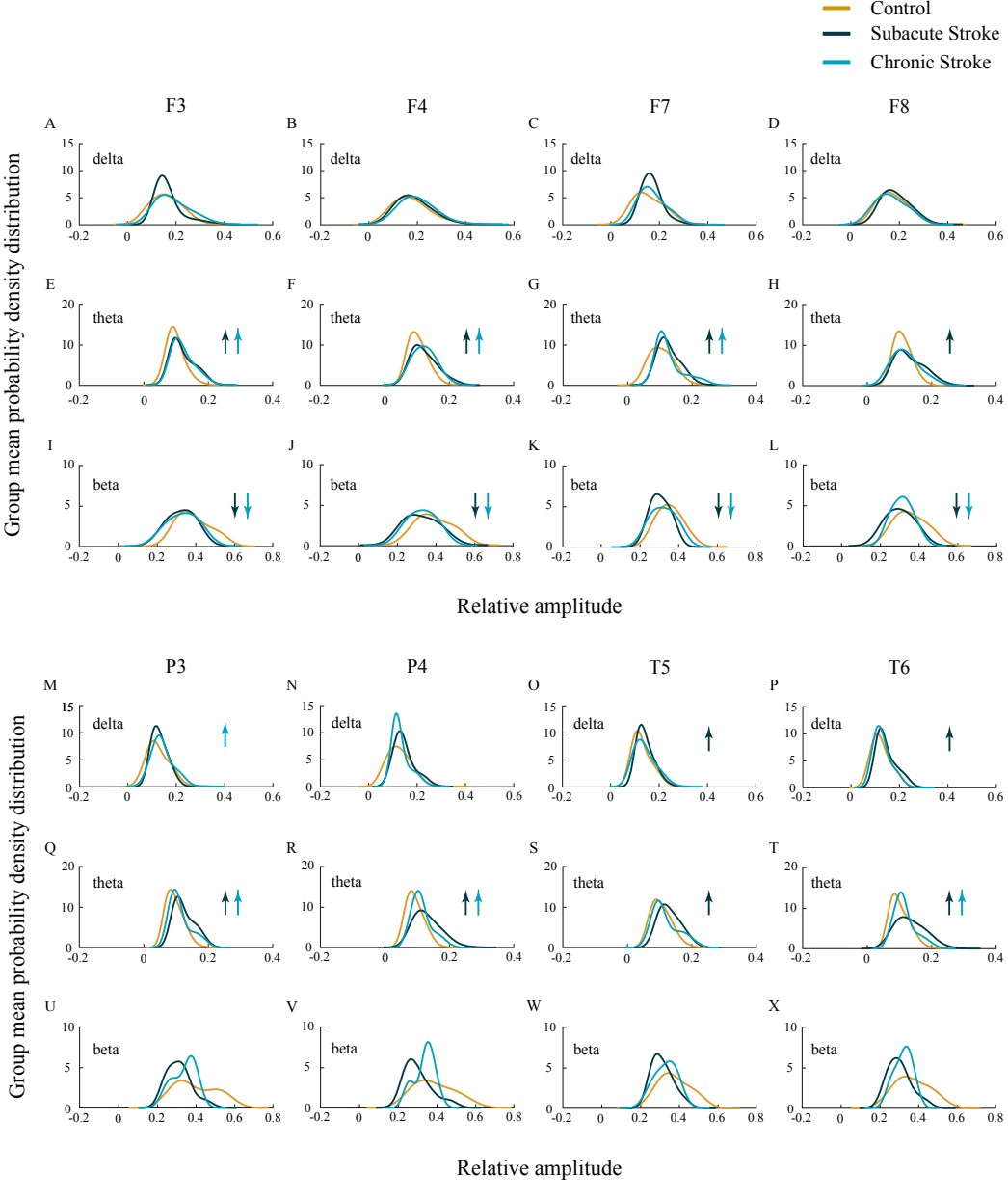

**Figure 2** **EEG microstructure following a minor stroke.** Group mean probability density distributions (PDEs) of the delta, theta and beta relative amplitudes at the frontal (A–L), parietal and temporal (B–X) EEG channels in the subacute ($n = 10$) and chronic ($n = 9$) stages of stroke patients vs. healthy controls ($n = 11$). The slower alpha in stroke patients with no amplitude change ($p \geq 0.14$) was followed by a generally augmented theta ($p \leq 0.03$), alongside an attenuated beta only within the frontal cortex and during both the subacute and chronic stage of the stroke ($p \leq 0.01$). These stroke induced theta and beta amplitude alterations were followed by a delta augmentation at T5 and T6 in the chronic stage. Arrows indicate the statistically significant EEG amplitude alterations in the corresponding group (the subacute or chronic stage of the stroke vs. the control).

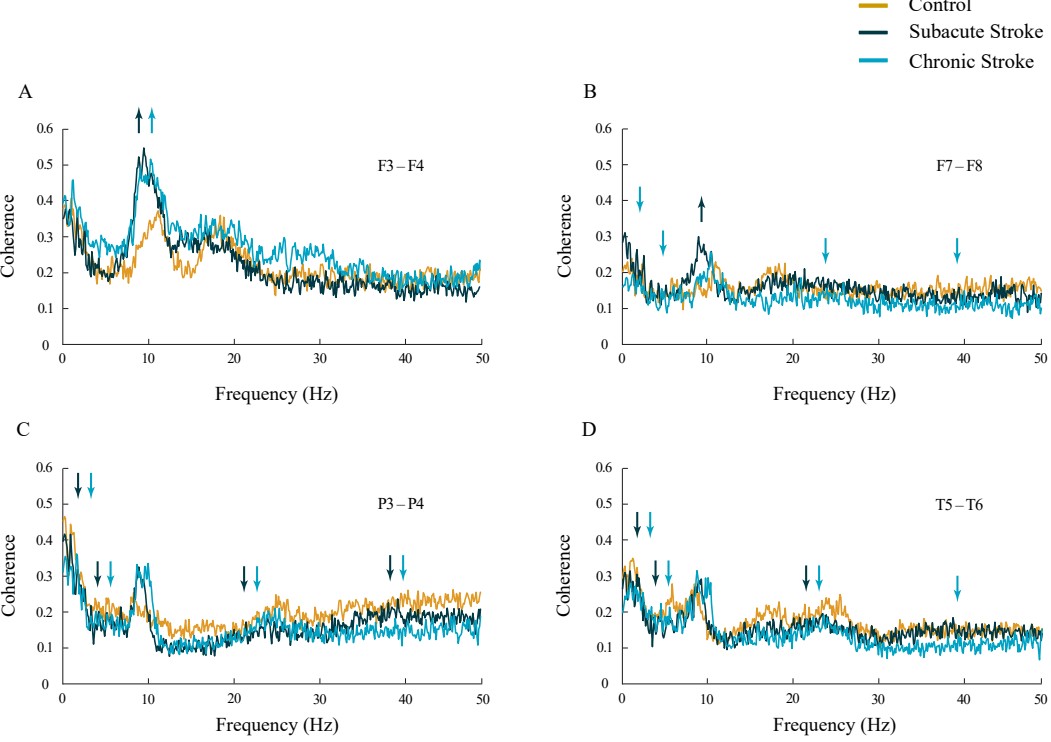

**Figure 3** **Inter-hemispheric coherences following a minor stroke.** The mean inter-hemispheric coherence spectra in the subacute ($n = 10$) and chronic ($n = 9$) stages of stroke patients vs. healthy controls ($n = 11$). In contrast to the parietal and temporal cortices (C, D) that expressed a generalized long-lasting desynchronization in all the frequency ranges ($p \leq 0.04$; C, D), but with no change in alpha coherence ($p \geq 0.10$), the frontal cortex exhibited an increased synchronization of slower alpha ($p \leq 0.02$) in stroke patients vs. controls (A, B). This increased synchronization of slower alpha was particularly expressed at F3–F4 electrodes (A), both in the subacute and the chronic stage. Arrows indicate the most consistent and significantly altered coherences in the corresponding group (the subacute or chronic stage of stroke vs. the control).

PP differences (Figs. 5A and 5B). This long-term impact of the stroke on the alpha PP differences was additionally confirmed by the almost non-existing PP differences between the subacute and chronic stage of the stroke patients (Fig. 5C). In addition, whereas the control group of patients showed relatively similar (large) alpha PPs (Figs. 5A and 5B, Control), the subacute stroke group of patients (Fig. 5A, Subacute Stroke) and the chronic stroke group of patients (Fig. 5B, Chronic Stroke) demonstrated a clear antero-posterior difference in alpha PPs, with the smallest frontal PPs vs. the largest posterior PPs. Consequently, the great majority of channels over the frontal cortex exhibited negative PP differences (dashed circles, Fig. 5A, Subacute Stroke vs. Control; Fig. 5B, Chronic Stroke vs. Control), meaning that the frontal alpha waves were more delayed in the stroke patients than in the controls compared to other channels. We identified the greatest decrease of PPs over the frontal cortex at F4 (Fig. 5A, Subacute Stroke vs. Control; Fig. 5B, Chronic

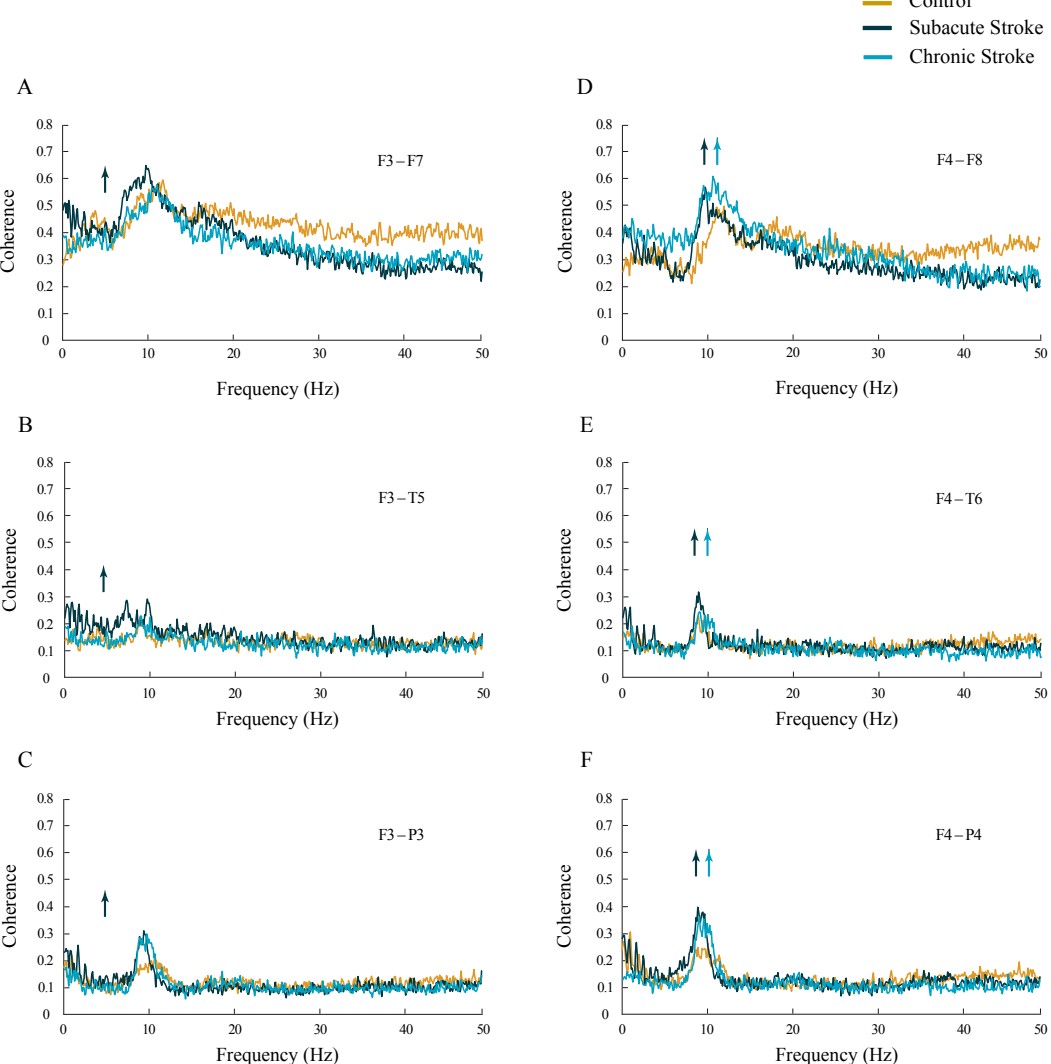

**Figure 4** **Intra-hemispheric coherences following a minor stroke.** The mean intra-hemispheric coherence spectra in the subacute ($n = 10$) and chronic ($n = 9$) stages of stroke patients vs. healthy controls ($n = 11$). In contrast to the increased synchronization of slower alpha in the ipsi-lesional hemisphere (D–F; $p \leq 0.05$) in the subacute and chronic stages of stroke patients vs. controls, the contra-lesional hemisphere exhibited a generalized but transient (only in the subacute stage) increase in theta synchronization (A–C; $p \leq 0.05$) with no alteration to alpha synchronization. Arrows indicate the most consistent and significantly altered coherences in the corresponding group (the subacute or chronic stage of stroke vs. the control).

Stroke vs. Control) and during both stages of stroke, which, as a sign of the largest delay in the newly emerged slower alpha, might be related to the confirmed neuropathology of an ischemic stroke.

Another way to present the group mean alpha PPs is to draw them as vectors in a unit circle (Figs. 6A–6C). In cases where the PPs of all channels are closely distributed in a unit circle, forming "a bundle", this formation could be considered as having a "source" (the channel with the highest PP, where the alpha wave is first recorded) or "drain" (the channel with the lowest PP, where the alpha wave is last recorded). Since all
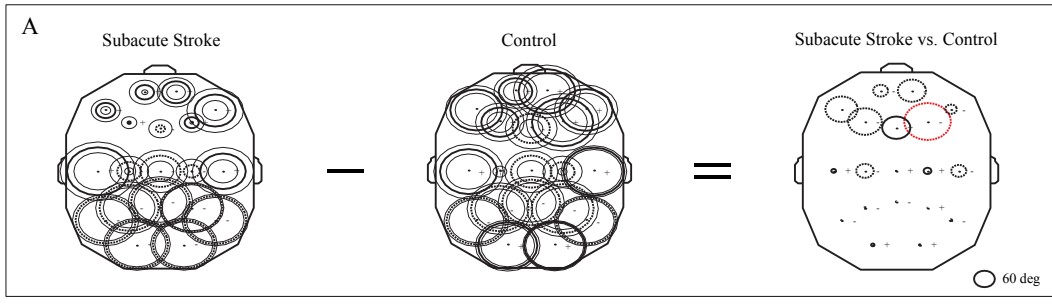

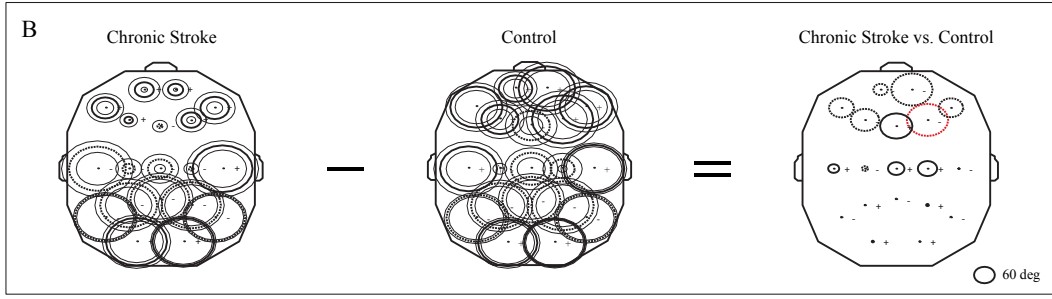

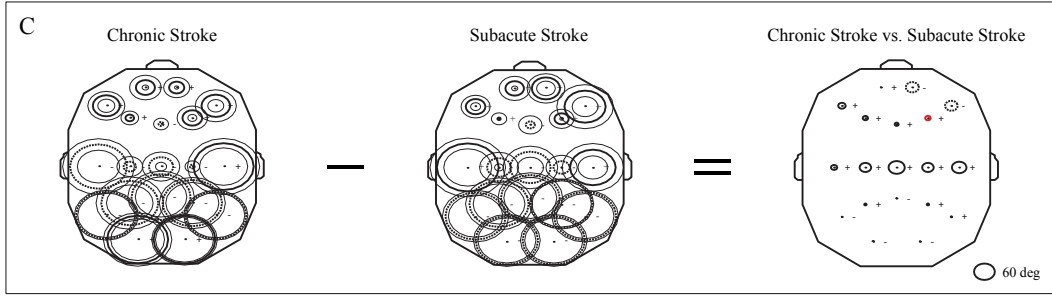

**Figure 5 Topographic distribution of alpha PPs following a minor stroke.** The group mean alpha PPs in the subacute ($n = 10$) and chronic ($n = 9$) stages of stroke patients vs. healthy controls ($n = 11$). We have shown that the frontal cortex generally exhibits the greatest PP differences related to a slower alpha than all the other EEG channels in both the subacute (A) and chronic stages of stroke patients (B) vs. the controls, particularly at F4 (the red dashed circle). The persistence of stroke induced alpha PPs differences was additionally confirmed by the almost non-existing PP differences between the subacute and chronic stages of the stroke patients (C). This result indicates that the newly emerged slower alpha in stroke patients might be related to the confirmed neuropathology of an ischemic stroke. Thick circles: the mean PPs; thin circles: the mean PPs ± SDs. Solid line: $0° < PP < 180°$; dashed: $-180° < PP < 0°$. Referential circle with $PP = 60°$, drawn in the lower right corner. According to the angular nature of PP, the large solid circles (e.g., $+175°$) have similar PP values to the large dashed circles (e.g., $-179°$).

the demonstrated alpha PPs were distributed across the whole circle, rather than being concentrated within "a bundle", the unit circle could be considered only as a closed loop or no "source" or "drain" unit circle. Therefore, we have drawn a topographic pattern of the "alpha flow" for the control group (Fig. 6D), the subacute stroke group (Fig. 6E), and the chronic stroke group (Fig. 6F) by connecting the neighboring PP EEG channels with arrows following the descending PP gradient. The greatest alterations in the "alpha flow" in stroke patients vs. the controls were demonstrated within the frontal cortex (Figs. 6G–6I,

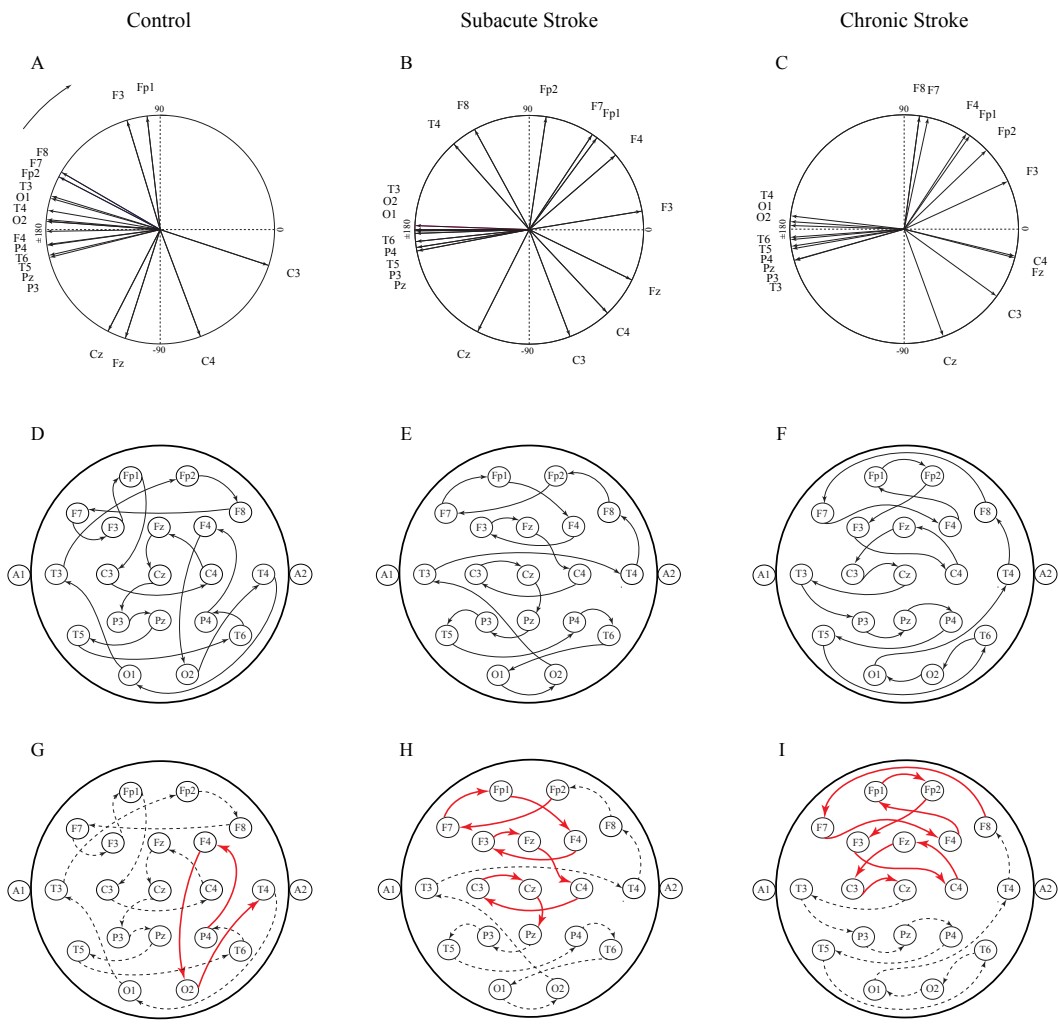

**Figure 6  Topographic pattern of "alpha flow" following a minor stroke.** A reconstruction of "alpha flow" in healthy controls ($n = 11$), the subacute ($n = 10$) and the chronic ($n = 9$) stages of stroke patients, based on the group mean alpha PPs distribution in the unit circle (A–C). The topography of the descending order of PPs is plotted by arrows connecting the channel with a higher PP to the channel with the next lower PP (D–F). The greatest alterations in stroke patients vs. the controls were expressed as a permanently established inter-hemispheric "alpha flow" (G–I).

indicated by solid red arrows), with F4 undergoing the most intensive PP change, leading to the reorganization of the PP channel sequence. To be specific, in control group the F4 PP neighbors were positioned longitudinally in the same ipsi-lateral hemisphere (Fig. 6G), whereas in the subacute (Fig. 6H) and chronic stage of the stroke patients (Fig. 6I) F4 was included in the newly established antero-posterior inter-hemispheric spiral PP pattern, indicating that the stroke had induced a sustained augmentation of inter-hemispheric interactions.

**Table 3** Correlations of MoCA and MoCA-MIS cognitive assessments and αAVG in the subacute and chronic stages of stroke patients vs. healty controls.

| | | MoCA vs. αAVG | | | | | | | |
| --- | --- | --- | --- | --- | --- | --- | --- | --- | --- |
| | | F3 | F4 | F7 | F8 | P3 | P4 | T5 | T6 |
| Control | rho | 0.06 | −0.02 | 0.13 | 0.10 | 0.01 | −0.20 | −0.23 | −0.24 |
| | p | 0.87 | 0.95 | 0.70 | 0.78 | 0.97 | 0.56 | 0.50 | 0.47 |
| Subacute Stroke | rho | 0.51 | **0.74** | 0.09 | 0.52 | **0.65** | **0.77** | 0.53 | **0.74** |
| | p | 0.13 | **0.02** | 0.82 | 0.12 | **0.04** | **0.01** | 0.12 | **0.02** |
| Chronic Stroke | rho | −0.04 | 0.34 | 0.08 | 0.39 | 0.13 | 0.15 | 0.22 | 0.19 |
| | p | 0.92 | 0.38 | 0.83 | 0.30 | 0.75 | 0.70 | 0.57 | 0.63 |

| | | MoCA-MIS vs. αAVG | | | | | | | |
| --- | --- | --- | --- | --- | --- | --- | --- | --- | --- |
| | | F3 | F4 | F7 | F8 | P3 | P4 | T5 | T6 |
| Control | rho | 0.23 | 0.14 | 0.30 | 0.21 | 0.20 | 0.00 | −0.02 | −0.04 |
| | p | 0.49 | 0.69 | 0.37 | 0.54 | 0.57 | 1.00 | 0.95 | 0.92 |
| Subacute Stroke | rho | 0.28 | 0.28 | 0.37 | 0.16 | 0.40 | 0.13 | 0.57 | 0.24 |
| | p | 0.44 | 0.43 | 0.30 | 0.66 | 0.25 | 0.71 | 0.09 | 0.51 |
| Chronic Stroke | rho | 0.56 | 0.63 | 0.61 | 0.52 | **0.72** | **0.92** | **0.77** | **0.92** |
| | p | 0.12 | 0.07 | 0.08 | 0.15 | **0.03** | **<0.01** | **0.02** | **<0.01** |

**Notes.**
Bold numbers indicate statistically significant correlations at $p \leq 0.05$.

## Correlations of the average EEG alpha frequency and cognitive functions

The results of correlation analysis between the MoCA and MoCA-MIS scores and the αAVG are presented in Table 3 and Fig. 7. We did not find a significant correlation between the MoCA and MoCA-MIS scores in the 11 control subjects and their αAVGs at any EEG channel ($p \geq 0.37$).

While the MoCA scores in our 10 subacute stroke patients were significantly correlated with the αAVGs at F4 ($\rho = 0.74$, $p = 0.02$), P3 ($\rho = 0.65$, $p = 0.04$), P4 ($\rho = 0.77$, $p = 0.01$) and T6 ($\rho = 0.74$, $p = 0.02$), the MoCA-MIS scores were not significantly correlated with the αAVGs at any EEG channel ($p \geq 0.09$).

Although the correlation between the MoCA scores and the αAVGs in our nine chronic stroke patients was not significant (they returned to the control correlation values, $p \geq 0.38$), the MoCA-MIS became significantly correlated with their αAVGs at P3 ($\rho = 0.72$, $p = 0.03$), P4 ($\rho = 0.92$, $p < 0.01$), T5 ($\rho = 0.77$, $p = 0.02$) and T6 ($\rho = 0.92$, $p < 0.01$).

## DISCUSSION

This study investigated EEG alpha activity and its relationship to post-stroke neuropathology, re-organizational processes, and cognitive functions during the subacute and chronic stage of minor stroke patients vs. healthy controls. We have demonstrated a generally slower alpha frequency (αAVG) in the subacute stroke patients vs. controls in a small sample of right MCA stroke patients (Fig. 1). This slower alpha involved no

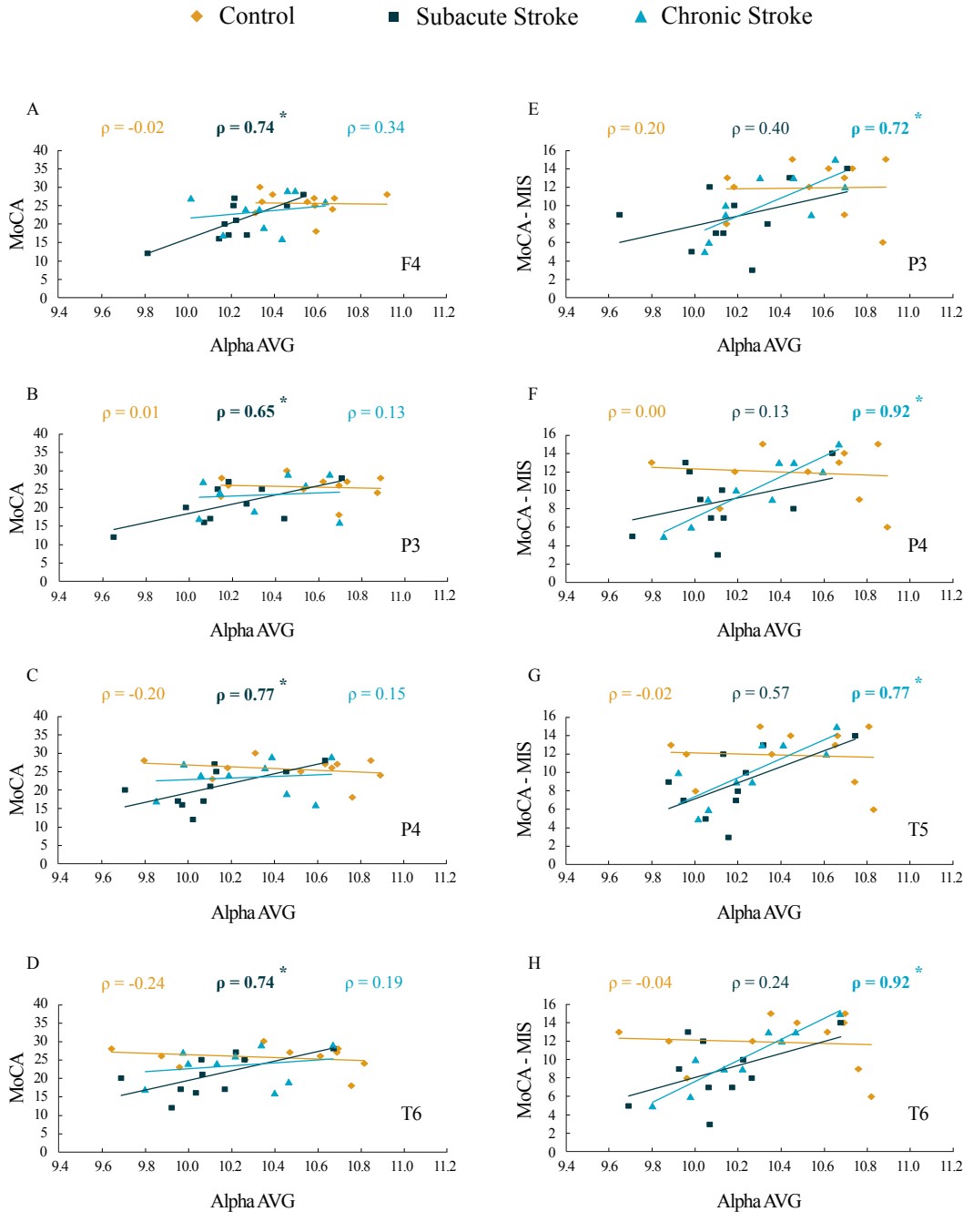

**Figure 7 Correlations of MoCA and MoCA-MIS cognitive assessments and $\alpha$AVG in the subacute and chronic stages of stroke patients vs. the controls.** The statistically significant correlations of $\alpha$AVG and MoCA (A–D) and MoCA-MIS (E–H) scores in the subacute and chronic stages of stroke patients vs. the controls. Bold numbers and asterisks indicate statistically significant correlations at $p \leq 0.04$.

amplitude change, but was highly synchronized intra-hemispherically, overlying the ipsi-lesional hemisphere (Fig. 4), and inter-hemispherically, overlying the frontal cortex (Fig. 3). In addition, the disturbances of EEG alpha activity in subacute stroke patients vs. controls were expressed as a decrease in alpha PPs over the frontal cortex (indicating the delay of the slower alpha), and an altered "alpha flow", indicating the sustained augmentation of inter-hemispheric interactions (Figs. 5 and 6). Although the stroke induced alpha frequency changes (the slower alpha) returned to the control values in chronic stroke patients, the increased alpha intra-hemispheric synchronization, overlying the ipsi-lesional hemisphere, the increased alpha F3–F4 inter-hemispheric synchronization, the delayed alpha waves, and the newly established inter-hemispheric "alpha flow" within the frontal cortex, remained as a permanent consequence of the minor stroke.

One way to consider the sustained alterations in EEG alpha activity in minor stroke patients is as a compensatory brain response contributing to functional recovery. It could be assumed that these re-organizational changes, expressed as a permanently established inter-hemispheric "alpha flow", may result from unbalanced inter-hemispheric inhibition, a phenomenon well known from transcranial magnetic stimulation (TMS) studies assessing motor function recovery after hemiparetic stroke (*Bütefisch et al., 2008*; *Manganotti et al., 2008*; *Shimizu et al., 2002*). To specify, normal brain activity depends on the balance between excitatory and inhibitory signaling, and following a stroke this balance is shifted toward excitation, leading to a disturbance in competitive inhibition between the two hemispheres, which is normally maintained during activation. The unilateral brain lesions reduce inhibition from the affected (ipsi-lesional) to the contralateral (contra-lesional) hemisphere, which in turn becomes hyperactive and potentiates the inhibition of the ipsi-lesional hemisphere (*Bütefisch et al., 2008*). In addition, evidence of brain hyperexcitability is provided by studies in animal models of stoke, demonstrating the down-regulation of the GABA receptor function and up-regulation of the NMDA receptor function (*Que et al., 1999*; *Redecker et al., 2002*). The increased excitability of the motor cortex and dynamical alterations to inter-hemispheric inhibition in post-stroke patients, demonstrated by the paired-pulse TMS technique, have been related to the degree of clinical motor recovery (*Bütefisch et al., 2008*; *Manganotti et al., 2008*).

Thus, in the context of post-stroke brain plasticity, we can only speculate that the newly established inter-hemispheric "alpha flow" in our study could be a way for the brain to compensate for the lesion and restore the lost function.

Ischemic brain injury results not only in a loss of local neural function but also in the disturbance of the remote functional networks. PET and fMRI imaging studies (*Carter et al., 2010*; *He et al., 2007*; *Warren et al., 2009*), investigating neural interactions between the brain regions in stroke patients, have demonstrated the disruption of the functional connectivity between the affected hemisphere and the rest of the brain. This observed impairment of neuronal interactions reflects neurological deficits (*Carter et al., 2010*; *He et al., 2007*; *Warren et al., 2009*). Moreover, MEG and EEG studies have shown the disruption of the functional connectivity of specific frequency bands, such as decreased alpha band synchronization (coherence) between the affected region and the rest of the brain (*Dubovik et al., 2012*; *Westlake et al., 2012*). The magnitude of alpha band synchronization was

negatively correlated with the functional impairment, with a distinct increase in alpha synchronization during recovery, and in parallel with the improvement of the neurological deficits (*Westlake et al., 2012*).

In our study, the functional connectivity between the brain regions was estimated using EEG coherence, particularly the mean coherence between the electrodes overlaying the area of the confirmed neuropathology. We have demonstrated increased alpha synchronization between the ipsi-lesional and contra-lesional frontal cortex (Fig. 3), and within the overall ipsi-lesional hemisphere (Fig. 4). This generalized alpha synchronization indicates the compensatory recruitment of peri-lesional as well as contra-lesional brain areas during post-stroke recovery.

We should mention here that the reference technique we used was determined by the manufacturer of the EEG apparatus (NK-9100K EEG system Nihon Kohden, Tokyo, Japan), and it was a limitation of our study. However, although the reference technique and volume conduction could affect coherence measures, we demonstrated an increased alpha synchronization, both inter-hemisphericaly, overlaying the frontal cortex (Fig. 3) and intra-hemisphericaly, overlaying the ipsi-lesional hemisphere (Fig. 4) in the stroke patients vs. the controls. In addition, our coherence analysis also included other frequency bands, and not all of them were increased in the stroke patients vs. the controls. Conversely, many of them exhibited desynchronization, i.e., decreased coherence (Figs. 3 and 4). If we had made a mistake, and our coherence measures had been affected by volume conduction, than the error would be incorporated in all the coherence measures, and all the experimental groups due to the reference impact. Despite that, we still showed the differences in alpha (and other frequency bands) coherences between stroke patients and healthy controls.

Our results of the cognitive assessments demonstrated the decreased MoCA and MoCA-MIS scores in stroke patients vs. the controls, but only during the subacute stage, indicating only a transient cognitive impairment due to a minor stroke. In parallel with the observed cognitive dysfunction, the stroke patients also expressed a transient decrease in αAVG (Fig. 1). In addition, our study revealed significant correlations between αAVG and the cognitive assessment scores in stroke patients vs. the controls (Table 3; Fig. 7). In contrast to the controls, who showed no correlations, the αAVGs in stroke patients were positively correlated with MoCA scores only during the subacute stage of the stroke, whereas MoCA-MIS scores became significantly correlated with αAVGs during the chronic stage of the stroke.

Generally, the greatest behavioral recovery following a stroke is shown within the first four weeks, but the functional recovery may continue beyond this initial period, particularly when restorative therapies are included during rehabilitation (*Green, 2003*; *Grefkes & Ward, 2014*; *Teasell, Bayona & Bitensky, 2005*). However, the rate of overall recovery depends on the severity of the stroke and the domain affected, with mild initial impairments showing faster improvement and better final outcomes in contrast to more severe strokes (*Cramer, 2008*). When it comes to cognitive impairment, most stroke survivors show their maximum recovery within the first three months (*Desmond et al., 1996*). Our results are in accordance with the concept of spontaneous functional recovery following a stroke. All the subjects in our study were assessed as minor stroke patients

(NIHSS < 4) with transient cognitive impairments that were demonstrated only during the subacute stage. Although the MoCA-MIS scores along with αAVGs returned to the control values in the chronic stroke patients, they became highly correlated (functionally coupled, Table 3, Fig. 7), only over the posterior brain region. This newly established functional coupling (topography) could be a compensatory reorganization following an ischemic lesion.

The functional relevance of alpha EEG oscillations has been extensively studied (*Klimesch, 1999*; *Klimesch, 2012*; *Palva & Palva, 2007*). A lot of evidence suggests that alpha oscillations reflect cognitive and memory performance (*Klimesch, 1999*). A study by *Patten et al. (2012)* has shown that the properties of alpha wave propagation over the scalp correlate with the speed of information processing, particularly for alpha waves traveling in the frontal-to-occipital direction. On the other hand, reaction time, as a measure of information processing speed, is correlated with alpha frequency: a lower alpha frequency means a slower reaction time, hence poor cognitive and memory performance (*Klimesch, 1999*). Moreover, the propagation of electrical oscillations can be affected by brain pathologies, such as tumors, ischemia or vasogenic edema (*Jochmann et al., 2011*). Based on these findings, we can speculate that the post-stroke cognitive impairment, as well as the alpha frequency decrease, demonstrated in our study, could be a consequence of the altered propagation of alpha waves due to a stroke induced lesion.

In addition, the alpha frequency is known to vary with age (increasing up to adulthood and decreasing in older age), age related diseases, or even a lack of school education (*Klimesch, 1999*). However, no statistically significant differences in age or education between the control and stroke group of patients and the lack of (other) neurological and psychiatric diseases in the stroke patients, indicates a stroke induced slower alpha frequency.

Apart from the EEG alpha activity, we have shown alterations to other frequency bands, and as expected, the stroke patients demonstrated pathological EEG slowing during both stages of a minor stroke (Fig. 2). Although, the stroke induced slowing of EEG is well documented (*Burghaus et al., 2013*; *Finnigan, Wong & Read, 2016*; *Jordan, 2004*), in our study the alpha slowing was followed by theta amplitude augmentation and beta amplitude attenuation, particularly over the frontal cortex. In addition, the minor stroke induced long-term alpha inter-hemispheric synchronization within the frontal cortex, followed by permanent delta, theta, beta, and gamma inter-hemispheric desynchronization within the parietal and temporal cortex (Fig. 3).

We have to note here that this study has some limitations. To be clear, we studied a small specific sample of stroke patients (nine stroke patients), all with minor strokes (NIHSS < 4) in the distribution of the right MCA. Although our findings could benefit from a larger sample size, we still can not assume that our results might be extrapolated to other types of stroke patients or even to left MCA stroke cases. Future research including other types of stroke patients (and a larger sample size) could resolve this issue and further prove or disprove the potential value of our EEG measurements as general qEEG indices.

Our study of a small sample of right MCA stroke patients has demonstrated EEG alpha activity alterations following a minor stroke. Whereas the transient slowing of the alpha

frequency was related to cognitive impairment in subacute stroke patients, the sustainably increased alpha synchronization, delayed alpha waves, and a newly established frontal inter-hemispheric "alpha flow" were permanent consequences. These permanent alpha alterations revealed the "hidden" stroke neuropathology or post-stroke re-organizational processes, despite the fact that the cognitive impairment returned to the control value. Our study indicates that slower EEG alpha generation, synchronization and "flow" may potentially serve as biomarkers of cognitive impairment onset and/or post-stroke compensatory re-organizational processes.

### Funding

This work was supported by the Serbian Ministry of Education, Science and Technological Development Grant OI 173022. The funders had no role in study design, data collection and analysis, decision to publish, or preparation of the manuscript.

### Grant Disclosures

The following grant information was disclosed by the authors:
Serbian Ministry of Education, Science and Technological Development: OI 173022.

### Competing Interests

The authors declare there are no competing interests.

### Author Contributions

- Jelena Petrovic and Vuk Milosevic conceived and designed the experiments, performed the experiments, analyzed the data, contributed reagents/materials/analysis tools, wrote the paper, prepared figures and/or tables, reviewed drafts of the paper.
- Miroslava Zivkovic performed the experiments, wrote the paper.
- Dragan Stojanov conceived and designed the experiments, analyzed the data, contributed reagents/materials/analysis tools, prepared figures and/or tables.
- Olga Milojkovic performed the experiments.
- Aleksandar Kalauzi conceived and designed the experiments, analyzed the data, contributed reagents/materials/analysis tools, wrote the paper, prepared figures and/or tables, reviewed drafts of the paper.
- Jasna Saponjic conceived and designed the experiments, analyzed the data, contributed reagents/materials/analysis tools, wrote the paper, reviewed drafts of the paper.

### Human Ethics

The following information was supplied relating to ethical approvals (i.e., approving body and any reference numbers):

The study was carried out in accordance with Declaration of Helsinki (The Code of Ethics of the World Medical Association), and approved by the Ethical Committee of the Medical Faculty, University of Nis, Republic of Serbia (No 12-9808-2/1), and all participants gave informed consent.

## Data Availability

The raw data has been provided as Supplemental Files.

## Supplemental Information

Supplemental information for this article can be found online at http://dx.doi.org/10.7717/peerj.3839#supplemental-information.

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
