# Peer review of "Slower EEG alpha generation, synchronization and “flow”—possible biomarkers of cognitive impairment and neuropathology of minor stroke"

_PeerJ, doi:10.7717/peerj.3839_

## Round 0.1 · original submission · Major Revisions

Your article has now been reviewed. Please note, as in the comments above, that it will be necessary for you to address all of the reviewer comments and suggested changes.

Reviewer 1 ·

Basic reporting

It is not immediately obvious in the abstract that stroke patients where tested at both sub-acute and chronic stages. Please add additional detail to make this clear.

The authors have used "ɑAVG" in equations for calculation of both weighted average alpha frequency (line 173) and probability density estimate (line 182). Is this an error?

Groups are represented by different shades of blue in all figures, and to my eyes, the light blue used for the control group is almost too faint to comfortably see. Consider using a different color scheme.

Experimental design

No comment

Validity of the findings

Age comparisons between groups were made for stroke patients in the sub-acute, but not chronic stage. Was the age of stroke patients at follow-up in the chronic stage still matched to controls? Potentially important given well described effects of advanced age on EEG alpha (as mentioned in the Discussion, lines 452–458).

Most significantly for the the coherence measures, it seems likely that volume conduction is a major biasing factor. Also a potential issue is the use of a common reference, in this case, the average of C3 and C4 (for example, see Nunez et al. 1997 Electroencephalogr Clin Neurophysiol 103:499–515; Nolte et al. 2004 Clin Neurophysiol 115:2292–2307). It is recommended the authors consider using analytical methods that limit these confounds (e.g. imaginary coherence).

Additional comments

No further comments.

Reviewer 2 ·

Basic reporting

LANGUAGE: Mostly good usage of English language throughout. Some errors e.g. “neurophysiological” wrongly used in place of “neuropsychological” (lines 88, 119); “above the overall” (twice, in line 368) should be changed to “overlying the”, or similar; “for the” is missing from “compensate for the lesion” (line 57 & elsewhere). The first sentence of Results in the Abstract (lines 43-46) is too long and complex, and needs to be separated into two sentences (at least).

INTRO, BACKGROUND, CONTEXT & LITERATURE: The Introduction provides a reasonable summary of literature relevant to this study, although with some exceptions and some shortcomings. In particular, the chief focus of this study is on EEG alpha activity. However the authors state in the Intro that “alpha and theta frequency could be of interest for the screening of post-stroke cognitive impairment” (line 116). Why then is theta activity not analyzed in more detail in this study? In addition the Schleiger et al (2014) study is cited, but the authors overlook the finding from that study that delta/alpha ratio (DAR) was found to correlate with post-stroke cognitive outcomes. Why was DAR not analyzed in this study? The authors also overlook that the Schleiger et al (2014) found alpha power from all electrodes (not frontal or posterior specifically) to correlate with post-stroke cognitive outcomes – but it seems that this study only analyzed frontal or posterior electrodes. The Intro and manuscript also is lacking a citation of Schleiger et al (2017) which reports findings that are more recent and relevant to this manuscript: including re peak alpha frequency and slowing of alpha after stroke. So more justification is needed in the Intro for the authors’ choice of EEG parameters. Or – more parameters (e.g. DAR, or lower-frequency / slower alpha range) could be analysed.
Line 83: post-stroke cognitive impairments are not always, and not only, consequences of frontal “impairments”/damage specifically. For example, damage to other brain regions – e.g. medial temporal, parietal cortex – can also results in cognitive impairments.

STRUCTURE: appears appropriate.
FIGURES: appear appropriate; possibly too many (7) figures?
RAW DATA: I’m not aware as to whether or not the raw EEG data are available?

Experimental design

SCOPE: appears to be within journal scope.

RESEARCH QUESTION: I think the question and primary aim is not very well defined or identified. We have stroke and control samples. From the stroke sample we have two assessment times – mean 9.7 days and mean 13.7 months after stroke. Was EEG recorded at both times? This is ambiguous (line 151). Stroke symptoms (by NIHSScale) and cognitive function (by MoCAssessment) were assessed at both times. It remains unclear as to whether the focus is on: controls v stroke? Longitudinal changes in stroke EEG? Correlations between (early) stroke EEG and outcomes (MoCA / NIHSS?) from stroke? Instead there seems an over-emphasis on EEG analyses and parameters. I believe the authors need to more clearly identify and state what is their primary research question and aim; also what is the secondary (and perhaps tertiary) question of aim.

STANDARDS OF INVESTIGATION: No issues (apart from lack of focus of primary question and aim).

METHODS DETAIL: Generally adequate, although a few missing details. Was EEG recorded at both assessment times? - this is ambiguous (line 151). Was (C3+C4)/2 the online, or offline, reference technique (or both)? In any case – this is unusual. Common average reference is mentioned in a later section (line 202), but was this applied for ALL EEG analyses and parameters? Was it average of all 19 electrodes used, or average of only the 8 electrodes analyzed? Were ONLY the 4 frontal and 4 posterior electrodes analysed (line 164)? – the authors say “particularly” but I think they mean “only”? Was relative power calculated relative to 1.1 – 50 Hz power, or other range (line 168-9)? For each frequency band was power summed or averaged? What was the frequency resolution of the FFT output?

Validity of the findings

Results seem inconclusive, mainly because they are from a small, specific sample of stroke patients: 9 patients with Right middle cerebral artery (RMCA) stroke, who had minor strokes (NIHSS < 5). This is acceptable for this journal, but this disclaimer needs to be stated. The “benefit”, or implications, for the literature could be proposed. For example – these findings from a (small) sample of RMCA patients, could not be extrapolated to LEFT MCA cases (even if the RMCA sample size was large).

DATA: the main issue is the small and specific nature of the sample (see above).

CONCLUSIONS: Need to be tempered (i.e. disclaimer added) as appropriate given that small, specific sample of stroke patients - 9 patients with Right middle cerebral artery (RMCA) stroke, who had minor strokes (NIHSS < 5) – were studied. Some conclusions or interpretations seems to general or perhaps speculative – e.g. re “alpha flow” (lines 375-7; 471; 59) and lines 449-451. Some speculation is ok (see below) but this should be presented as speculative proposal(s) rather than “firm, conclusive or final”.
Discussion paragraph (lines 452-8) seems quite irrelevant and could be deleted (or shortened).

SPECULATION: The conclusions and claims of the authors, as they stand, need disclaimers or caveats (as recommended above) and thus should be considered relatively speculative at this early stage.

Additional comments

I encourage the authors to revise the manuscript as recommended - this will result in an improved version of what is a quite interesting study and manuscript.

---

## Round 0.2 · Minor Revisions

Please ensure that you make all the necessary remaining revisions as advised by the reviewers.

In addition, I strongly recommend that the manuscript is reviewed carefully (ideally by a native English speaker if possible) in order to improve the English and eliminate grammatical errors. This will increase the likelihood of acceptance, but this is still not guaranteed.

For example, the definite article 'the' appears excessively throughout the manuscript - including an unnecessary 'the' in the title of the manuscript before the word 'possible'.

Reviewer 1 ·

Basic reporting

No comments

Experimental design

No comments

Validity of the findings

With regard to volume conduction and use of a common reference biasing coherence measures: I am willing to accept that the findings of this study are still valid without altering the analyses. Nevertheless, the potential influence of these confounds is still a limitation that needs to at least be acknowledged in the Discussion.

Additional comments

Nil

Reviewer 2 ·

Basic reporting

Reporting is generally satisfactory. I recommend just two improvements:
[1] The second-last sentence of the Discussion (lines 499-504, tracked version) is simply too long, and thus it also becomes potentially confusing. Please break this down into at least two sentences.
[2] The revised Aim at end of Intro (lines 141-145) is improved, however it remains too long and thus, the aims seem to diffuse or lacking adequate focused. Please deconstruct this into at least two sentences - the first should contain the principal/ primary aim, and the second sentence can state secondary/ supplementary aims.

Experimental design

No further comments.

Validity of the findings

No further comments.

Additional comments

I feel the authors have done a reasonable job in revising their manuscript in response to comments from myself and the other reviewer. If the two very long sentences (at end of Intro and end of Discussion) can be revised and improved (following my comments above) i believe this manuscript will then be suitable for publication. I wish the authors well in their future research in this field.

---

## Round 0.3 · accepted · Accept

Congratulations on the acceptance of your manuscript.